# Luteinizing Hormone Receptor Is Expressed in Testicular Germ Cell Tumors: Possible Implications for Tumor Growth and Prognosis

**DOI:** 10.3390/cancers12061358

**Published:** 2020-05-26

**Authors:** Mette Lorenzen, John Erik Nielsen, Christine Hjorth Andreassen, Anders Juul, Birgitte Grønkær Toft, Ewa Rajpert-De Meyts, Gedske Daugaard, Martin Blomberg Jensen

**Affiliations:** 1Group of Skeletal, Mineral and Gonadal Endocrinology, Department of Growth and Reproduction, Rigshospitalet, University of Copenhagen, 2100 Copenhagen, Denmark; mette.lorenzen.01@regionh.dk (M.L.); christine.hjorth.andreassen@regionh.dk (C.H.A.); 2Department of Growth and Reproduction, Rigshospitalet, University of Copenhagen, 2100 Copenhagen, Denmark; john.erik.nielsen@regionh.dk (J.E.N.); Anders.Juul@regionh.dk (A.J.); Ewa.Rajpert-De.Meyts@regionh.dk (E.R.-D.M.); 3Department of Pathology, Rigshospitalet, University of Copenhagen, 2100 Copenhagen, Denmark; Birgitte.Groenkaer.Toft@regionh.dk; 4Department of Oncology, Rigshospitalet, University of Copenhagen, 2100 Copenhagen, Denmark; kirsten.gedske.daugaard@regionh.dk; 5Division of Bone and Mineral Research, Harvard School of Dental Medicine, Harvard University, Boston, MA 02115, USA

**Keywords:** testicular germ cell tumor, testicular cancer, luteinizing hormone/choriogonadotropin receptor, luteinizing hormone, human chorionic gonadotropin

## Abstract

Luteinizing hormone/choriogonadotropin receptor (LHCGR) regulates gonadal testosterone production and recent studies have suggested a growth-regulatory role in somatic cancers. Here, we established that LHCGR is expressed in a fraction of seminoma cells and germ cell neoplasia in situ (GCNIS), and the seminoma-derived cell line TCam2 released LHCGR into the medium. LH treatment induced proliferation of TCam2 cells in vitro, while hCG treatment induced a non-significant 51% increase in volume of tumors formed in a TCam2 xenograft model. A specific ELISA was used to detect a soluble LHCGR in serum. Serum concentrations of soluble LHCGR could not distinguish 4 patients with GCNIS and 216 patients with testicular germ cell tumors (TGCTs) from 297 infertile or 148 healthy young men. Instead, serum LHCGR levels were significantly higher in 112 patients with a seminoma >5 cm or elevated serum lactate dehydrogenase (LDH) compared with men harboring smaller seminomas <2 cm or normal LDH levels. Serum LHCGR levels in TGCT patients could not predict relapse irrespective whether determined pre- or post-orchiectomy. Combined, these novel findings suggest that LHCGR may be directly involved in the progression and growth of seminomas, and our retrospective pilot study suggests that serum LHCGR may have some prognostic value in men with seminoma.

## 1. Introduction

Testicular germ cell tumors (TGCTs) represent one of the most common solid tumors in young men [1]. TGCTs of young adult type originate from a precursor cell, germ cell neoplasia in situ (GCNIS), that retains fetal germ cell characteristics, but undergoes malignant transformation after puberty and forms either an invasive seminoma or a non-seminoma [2]. Transition from GCNIS to the more invasive tumors is poorly understood but chromosomal aberrations and endocrine changes are important. This is illustrated by the abrupt rise in TGCT incidence after the pubertal increase in luteinizing hormone (LH), follicle stimulating hormone (FSH), growth hormone, and sex hormone production [3]. Treatment of TGCTs has a remarkably high cure rate, but treatment of advanced disease with chemotherapy is associated with acute and chronic adverse effects that in these young patients may impair quality of life [4]. 

A search for novel biomarkers for seminomas is particularly warranted because the existing clinical markers such as human chorionic gonadotropin (hCG) and/or α-fetoprotein (AFP) are used primarily as reliable biomarkers and proxy for the presence of non-seminomas [5]. Seminoma is associated with elevated levels of lactate dehydrogenase (LDH) but the prognostic potential of this marker is not without flaws since LDH may be elevated in various other non-malignant conditions [6]. The gonadotropic hormones: FSH, LH, and hCG are essential for sex steroid production, testis development, and spermatogenesis. LH is produced by the pituitary gland and stimulates testosterone production through activation of the luteinizing hormone/choriogonadotropin receptor (LHCGR) in Leydig cells [7,8]. hCG is undetectable in circulation in adulthood but stimulates testosterone production during the male masculinization window in the first trimester of human pregnancy, whereas testicular testosterone production in the second and third trimester is driven by pituitary LH secretion. LHCGR is a classical G-protein-coupled receptor expressed in Leydig cells and mediates all actions of LH and hCG. The LHCGR gene contains 11 exons that can be divided into three well-defined domains: a long, heavily glycosylated N-terminal extracellular ligand binding domain; a transmembrane domain; and a short C-terminal signaling tail. In addition to full-length LHCGR, several truncated variants are produced by alternative splicing [9]. Some of these isoforms have been suggested to be secreted and LHCGR may also be cleaved off leading to a cell free soluble LHCGR that has been shown in serum and follicular fluid of women [10] and serum of men [11]. Several studies have suggested that LHCGR may be involved in the pathogenesis of endocrine related cancers—for instance breast [12], ovarian, and endometrial [13], adrenocortical [14], pancreatic [15] and prostate [16] cancer. The remarkable finding of GCNIS in a patient with an activating LHCGR mutation supported a putative causal link between LHCGR and TGCT [17]. Here, we investigated the expression of LHCGR in GCNIS, TGCTs, and TGCT-derived cells lines and the effects of LHCGR activation on proliferation of testicular cancer cell lines in vitro and in two tumor xenograft mouse models. The prognostic potential of LHCGR was evaluated by investigating LHCGR levels in serum of men with TGCTs.

## 2. Results

### 2.1. Expression of LHCGR Isoforms in Testicular Germ Cell Tumors

RT-PCR using a primer set targeting exon 11 coding the intracellular and transmembrane region resulted in one clear band that was sequenced and confirmed to be LHCGR. This transcript was present in normal testis with spermatogenesis (NT), seminoma (SEM), and the seminoma-derived cell line TCam2 but not in the embryonal carcinoma-derived cell line NTera2. The primer set targeting exons 2–4 coding the extracellular region presented with multiple bands and two of these bands (indicated by red arrows) were confirmed to be LHCGR by direct sequencing. These transcripts were present in NT, GCNIS, and seminoma samples (Figure 1A). qPCR showed expression of LHCGR in NT, GCNIS, and in specimens with seminoma (Figure 1B). Presence of LHCGR in GCNIS and seminoma was verified by western blot using three different antibodies targeting different parts of the LHCGR. The antibodies targeting the C-terminal and an internal region revealed similar results with a clear band around 50 kDa in all tissue samples including embryonal carcinoma (EC) and SEM but expression seemed stronger in NT and GCNIS samples that also contained Leydig cells (Figure 1C). The antibody targeting the extracellular region also presented with a band around 50 kDa in all tissue samples, but this band seemed to be slightly above the bands found with the other antibodies. Staining with the antibody targeting the C-terminal of LHCGR resulted in additional bands, with the strongest band having a MW of about 75 kDa in all investigated specimens. This band was not detectable with the other investigated antibodies in any of the tissues including testis, while the antibody raised against an internal region was undetectable in both cell lines (Figure 1C and Table 1).

### 2.2. LHCGR Protein Detected in GCNIS and Seminoma

The monoclonal antibody LHR029, used for WB, and the polyclonal antibody LHRsc both targeting the extracellular domain of the LHCGR (amino acid residues 229–291 and 28–77, respectively) showed presence of LHCGR in normal testis, GCNIS and TGCTs. Interestingly, most GCNIS cells did not express LHCGR, however, staining with LHRsc revealed localization of LHCGR in the membrane of a subset of GCNIS cells (Figure 1D and Table 1). In accordance, LHCGR was also detected in 0–40% of the D2-40 positive seminoma cells. LHRsc revealed expression of LHCGR in the membrane of seminoma cells, whereas the LHR029 antibody presented with moderate to strong cytoplasmic staining of the seminoma cells. LHR029 also detected LHCGR in the cytoplasm of a subset of SOX2 positive EC cells while LHRsc showed no expression in non-seminoma components (Figure 1D). Staining of the seminoma-derived TCam2 cells with LHR029 and LHRsc showed a cytoplasmic/membranous expression of LHCGR. Expression of LHCGR was also investigated in a patient with an activating mutation in LHCGR. A testicular biopsy was taken when the patient was 1.5 years old showing Leydig cell hyperplasia with early maturation of some Leydig cells. At 25 years of age, an ultrasound revealed a testicular tumor, which proved to be a Leydig cell adenoma, but unexpectedly with GCNIS present in adjacent seminiferous tubules [17]. Interestingly, staining of specimens with LHRsc revealed expression of LHCGR in the Leydig cells but also in the cytoplasm of some of the GCNIS cells (Figure 1E and Table 1).

### 2.3. LHCGR Activation Regulates Proliferation of a Seminoma-Derived Cell Line In Vitro and In Vivo

LHCGR was detected in the medium from cultured TCam2 but not from NTera2 cells (Figure 2A), which is in accordance with the higher mRNA expression of LHCGR in TCam2 cells (Figure 1A). The seminoma-derived cell line TCam2 was treated with LH or hCG for 6, 24, and 48 h, and a significant increase in proliferation was observed after 24 (∆14%, *p* = 0.004) and 48 h (∆12%, *p* = 0.009) following treatment with LH compared with vehicle (Figure 2B,C). No effects of LH and hCG were observed in the NTera2 cell line. Subsequently, tumor growth was assessed following treatment with LH (3 IE/kg 5 times/week) or hCG (1000 IE/kg 3 times/week) compared with vehicle (PBS) or cisplatin (6 mg/kg once) in NTera2 or TCam2 xenograft mice models. All mice developed tumors during the study period. NTera2 tumors grew quickly and mice were therefore sacrificed at day 27. TCam2 tumors had a much slower growth and mice were sacrificed at day 49. hCG-treated tumors tended to grow faster than those treated with vehicle in both models. TCam2 tumors treated with hCG for 49 days were on average 51% larger than vehicle-treated tumors. However, the difference was not statistically significant due to large variability. Treatment with LH had no effect on tumor growth in any of the models. Cisplatin-treated mice had a significantly reduced tumor growth compared with vehicle-treated mice exclusively in the NTera2 xenograft model (Figure 2D,E).

### 2.4. Serum LHCGR Is Associated with Tumor Burden and Elevated LDH in Seminoma Patients

Presence of LHCGR in human serum was supported by conducting western blot on serum from two healthy men (with a low and high serum LHCGR level) and two patients with either seminoma or non-seminoma and high serum LHCGR levels measured by a specific ELISA platform. WB showed two bands at 50 and 75 kDa in the albumin and immunoglobulin depleted serum samples with all three antibodies, and thus of similar size as the bands detected in human testis (Figure 3A). The used LHCGR ELISA showed that young healthy men had the highest concentration of serum LHCGR, whereas infertile men or patients with benign tumors had lower serum LHCGR levels. Men with seminoma and non-seminoma tended to have higher serum LHCGR levels compared with men with GCNIS although they were not significantly different (Figure 3B). To evaluate the clinical and prognostic potential of LHCGR, patients with TGCTs (*n* = 216) were stratified according to tumor histology and subsequently sub-grouped according to serum levels of routinely used tumor markers: hCG, LDH, and AFP. In 112 patients with seminoma, serum LHCGR levels were associated with tumor size; patients with large tumors above 5 cm in diameter had higher serum concentration of LHCGR than men with small tumors <2 cm (*p* = 0.04) (Figure 4A). LDH is a non-specific marker for seminoma and men with elevated LDH (>205 U/L) also had higher serum LHCGR than seminoma patients with normal LDH levels (*p* = 0.009). No difference in serum LHCGR was found between patients with normal or elevated hCG levels (>3 IU/L) (Figure 4B). The risk of relapse in seminoma patients was associated with tumor size (*p* = 0.03) (Figure 4C), high LDH levels (*p* = 0.002) (Figure 4F), while serum LHCGR and hCG were not significantly different between patients with and without relapse (Figure 4D,E). In patients with non-seminomas prior to orchiectomy, serum LHCGR was not associated with tumor size (Figure 4G), but it was higher in men with elevated AFP (>12 × 10 ³ IU/L) (*p* = 0.026) (Figure 4H). Risk of relapse in patients with non-seminomas was not associated with tumor size or LHCGR (Figure 4I,J), but patients experiencing relapse had higher serum level of hCG (*p* = 0.0097) and AFP (*p* = 0.025) at the time of diagnosis (Figure 4K,L).

### 2.5. Evaluation of Serum Level of LHCGR during Monitoring of TGCT Patients with and without Relapse

The diagnostic value of serum LHCGR for detection of relapse was investigated in 23 patients with non-seminoma and 20 patients with seminoma followed longitudinally after orchiectomy. All included patients had stage 1 disease and were not treated with chemotherapy or any other adjuvant treatment. Blood samples were taken at the first control visit after orchiectomy and continuously at every visit during the follow-up period. Nine of 20 included patients with seminoma experienced a relapse and the average time from orchiectomy to relapse was 701 days (median: 384 days). Among the patients with non-seminoma, 11 out of 23 had a relapse and the mean time from orchiectomy to relapse was 253 days (median: 157 days). Serum LHCGR levels were not different between patients with and without relapse for both types of TGCTs at all the investigated timepoints (Figure 5A–D). There was a marked inter-individual variability in LHCGR serum levels, but the patients maintained either low or high levels throughout the follow-up period.

## 3. Discussion

This study provides novel evidence for a non-classical role of LHCGR in transformed germ cells. A prerequisite for functional relevance is presence of the receptor, which was verified by showing expression of LHCGR at mRNA and protein level in some GCNIS and seminoma specimens from TGCT patients. 

Previous studies have shown that LHCGR exists in various isoforms and some of them may even be secreted. The direct comparison of a primer set targeting exon 2–4 (the extracellular region) with a primer set targeting exon 11 (intracellular and transmembrane region) was performed to determine whether a truncated LHCGR isoform without the transmembrane/intracellular domain could be present. Presence of exon 2–4 and exon 11 in both seminoma and some GCNIS specimens suggests that LHCGR is available either as full-length or at least possesses the ability to signal intracellularly. The functional significance of all the LHCGR isoforms in seminoma remains to be established, but the presence of additional likely unspecific signals with the primer set targeting exon 2–4 highlights that caution is warranted during assessment of LHCGR expression. qPCR of a larger number of specimens corroborated the presence of LHCGR in seminoma and GCNIS specimens, while non-seminomas had low LHCGR or undetectable expression. The expression of LHCGR in GCNIS was higher with qPCR than observed with RT-PCR, which may be explained by cellularity due to a higher GCNIS abundance in the specimens and/or normalization to a housekeeping gene. The observed differences in gene expression levels between the samples were not evident at the protein level as few GCNIS cells in each specimen expressed LHCGR. 

LHCGR antibodies have a poor reputation and here we tested three different antibodies that all showed a similar band size of LHCGR around 50 kDa in normal testis, GCNIS, seminoma, and EC. Bands of similar size have been found in other studies investigating LHCGR in human tissues [18,19]. However, only the antibody targeting the C-terminal of LHCGR showed the predicted molecular weight of full length LHCGR around 75 kDa, which was detectable in all investigated tissues with the weakest expression in the normal testis. Instead, in normal testis and GCNIS samples that also contain Leydig cells, the intensity of the 50 kDa band was strongest with the antibody LHR029, which seemed to provide the most robust expression and was also used for IHC and the ELISA assay. Two different antibodies targeting the extracellular region showed expression of LHCGR in seminoma cells by IHC although with different subcellular localizations. This may be due to different epitopes but indicates that LHR029 and LHRsc both may be suitable for IHC analysis of LHCGR in human tissues. Interestingly, both antibodies also showed a weak cytoplasmic staining in GCNIS cells from a patient with testotoxicosis and thus supporting a direct role for LH/hCG in GCNIS and seminoma cells rather than solely indirectly through sex steroids.

To investigate the effect of LHCGR in seminomas, the seminoma-derived TCam2 cells, which had a cytoplasmic/membranous expression of LHCGR, were exposed to LH treatment, which increased proliferation. This suggested a growth regulatory role, which only partly could be corroborated in vivo using treatments with LHCGR agonists. We used a TCam2 xenograft mouse model, which showed no effect of LH and a large variability in the hCG-induced tumor size. Nevertheless, the observed average 51% increase in tumor volume in the hCG treated mice compared with vehicle treatment could be of relevance despite of not reaching statistical significance. Especially since the tumor growth reducing effect of cisplatin treatment also did not reach statistical significance compared with vehicle treatment, which indicates that even clinically relevant effects may be underestimated. The selected doses of LH and hCG in the in vitro and in vivo studies were based on previous studies investigating these drugs in cancer cell lines [20,21] and xenograft mouse models [21,22]. However, the use of nonequivalent doses of LH and hCG may explain the lack of effect of LH in vivo. The selected LH dose was three-fold higher than the normal human dosage, while the hCG dosing is difficult to extrapolate clinically because it is exclusively used for initiation of spermatogenesis in men with hypogonadotropic hypogonadism. Another possible explanation is that the two human agonists may have different potencies in mice or activate different signaling pathways downstream of the LHCGR as reported previously [23,24]. The proposed growth stimulatory response in the seminoma is in accordance with the reported stimulatory effect of hCG in somatic cancers [25,26,27,28,29,30]. Moreover, presence of sporadic GCNIS cells in the patient harboring an activating mutation in LHCGR supports a direct role for LHCGR in the pathogenesis of TGCTs. In accordance, another study showed that a patient with an activating mutation in LHCGR developed a seminoma, which supports a direct link between GCNIS/seminoma risk and LHCGR [17,31]. In contrast, there was no consistent expression of LHCGR in non-seminoma components and no growth regulatory effects of LH or hCG were found on NTera2 cells in vitro or in vivo, suggesting that seminoma and GCNIS appear to be the main targets for LHCGR agonists. 

Detection of LHCGR in the medium from TCam2 cells indicates that seminoma cells in patients also may release LHCGR. Presence of a factor in human serum, which can bind to LH and hCG has previously been shown [32] but soluble LHCGR was first purified from porcine follicular fluid in 1986 [33]. Later, it was identified in the medium of ligand-induced rat and mouse Leydig cells [34] or human cells transfected with cloned LHCGR [18,35]. In humans, LHCGR has been found in serum and follicular fluid of women [10] and by proteomics in serum from men and women [36] but could it also be released from seminoma cells and be measured in serum? We have recently shown that different LHCGR fragments are released into serum of healthy men [11]. The protein identified in the current study of around 75 kDa in serum from men with TGCTs may be representing the full-length receptor, while the lower molecular weight band (50 kDa) may represent a truncated or cleaved receptor protein. This suggestion is supported by showing similar band sizes in gonadal tissues and is in accordance with previous studies investigating LHCGR in serum [10,35,36]. We hypothesized that the circulating LHCGR or its fragment could have diagnostic potential in patients with TGCT. However, the study of LHCGR during clinical follow-up of TGCT patients showed that LHCGR had no diagnostic potential. LHCGR was unable to distinguish men with TGCTs from healthy men or men with benign testicular lesions. However, the higher serum LHCGR in patients with large seminomas compared with patients having small tumor volumes suggests that serum LHCGR may have some prognostic value in seminoma patients. Moreover, this finding implies that LHCGR may be released from the seminoma cells in vivo as shown for TCam2 cells in vitro. The link with tumor burden was corroborated by showing that serum LHCGR was higher in seminoma patients with elevated serum LDH, a marker which has previously been associated with more advanced disease and larger tumor burden [37]. The link between LHCGR and tumor burden is of clinical interest but the clinical use of LHCGR may be questioned if LHCGR adds no additional clinical value to the unspecific LDH measurements. The observed link with tumor volume and LDH was not found in patients with non-seminomas. Instead, LHCGR levels were significantly higher in patients with elevated AFP, which indicates that LHCGR in non-seminoma either originates or is related to the yolk sac tumor component [38]. The longitudinal part of our study showed that LHCGR cannot be used as a predictive marker during active surveillance after orchiectomy. Instead, the best predictors of relapse in patients with stage 1 seminoma were serum LDH and tumor volume. Serum LHCGR was not significantly different between patients with and without relapse in both seminoma and non-seminoma patients. 

## 4. Materials and Methods

### 4.1. Human Tissue Samples

Human adult testis and TGCT samples were collected from patients undergoing orchiectomy. Samples were collected in accordance with the Helsinki declaration after approval from the local ethics committee (permit no. H-1-2012-007). Each sample was divided and either snap frozen and stored at −80 °C or fixed in formalin, Bouin’s, or in modified Stieve’s fixative at 4 °C overnight followed by paraffin embedding. Each of the samples were carefully evaluated by an experienced pathologist and specific markers were used to determine the subtype of TGCT. In addition, testis samples retrieved after orchiectomy from a patient with an activating mutation in LHCGR was also included [17]. The included GCNIS samples always contain some other cellular elements, including Leydig cells and tubules with ongoing spermatogenesis. In most cases, each TGCT sample was studied individually by only one of the performed techniques.

### 4.2. Cell Culture, qPCR, and Proliferation Assay

All cell experiments were carried out using the embryonal carcinoma-derived cell line NTera2 [39] and the seminoma-derived cell line TCam2 [40]. NTera2 cells were grown in Dulbecco’s modified Eagles medium (DMEM) and TCam2 cells in Roswell Park Memorial Institute (RPMI) medium both supplemented with 10% fetal bovine serum (FBS), 100 U/mL penicillin, 100 mg/mL streptomycin and 58.6 mg/mL glutamine (all reagents purchased from Invitrogen, Carlsbad, CA, USA) and maintained in an incubator at 37 °C with 5% CO_2_ and the medium was changed every second day. ‘Normal testis’ RNA was purchased from three different companies (Clontech, Takara Bio Europe, Paris, France; BioChain Institute, Newark, CA, USA; Ambion, Life Technologies, Thermo Fisher Scientific, Waltham, MA, USA) and ‘normal ovary’ RNA was purchased from Abcam Plc, Cambridge, UK. RNA purification from tissues and cell lines, cDNA synthesis, and RT-PCR were performed as previously described [41]. cDNA was synthesized using 1 µg of RNA. Gene expression levels were analyzed using quantitative PCR on a Quantstudio 3 Real-Time PCR System. Samples were run in technical triplicates on a 96-well plate. Each reaction contained: 1 μL of cDNA, 1 μL of forward and reverse primer (10 pmol/μL), 12 μL H2O and 15 μL Brilliant II SYBR^®^ Green QPCR Master Mix (Agilent, Santa Clara, CA, USA, cat. no. 600828). PCR conditions were: 15 min at 95 °C followed by 40 cycles of 15 s at 95 °C, 1 min at 62 °C and a subsequent melting curve analysis. Changes in gene expression were quantified using the 2^−ΔΔCt^ method with RPS20 as internal control. The following primers were used: RPS2O (fwd: 5′-AGACTTTGAGAATCACTACAAGA-3′, rev: 5′-ATCTGCAATGGTGACTTCCAC-3′), LHCGR exon 2–4: (fwd: 5′-CCTACCTCCCTGTCAAAGTG-3′, rev: 5′-ATGCTCCGGGCTCAATGTATC-3′), and LHCGR exon 11 (fwd: 5′-CGATTTCACCTGCATGGCAC-3′, rev: 5′-GTGTAGCGAGTCTTGTCTAG-3′). A cell proliferation BrdU assay (Sigma, St. Louis, MO, USA, cat. no. 11669915001) was used to analyze proliferation according to the manufacture’s protocol. Briefly, cells were incubated in the presence of the test substance in a 96-well microplate in a final volume of 100 µL/well for 6, 24, and 48 h. 4 h before the end of each treatment period, 10 µL of BrdU labeling solution was added to each well and the cells were re-incubated at 37 °C. After 4 h the labeling medium was removed, and the cells were fixed and their DNA was denatured by addition of 200 µL of FixDenat. Next, cells were incubated with a BrdU antibody for 90 min at RT. Cells were washed three times in PBS before addition of a substrate solution and light emission was measured using a scanning multi-well spectrophotometer at 370 nm with a reference wavelength of 492 nm. Treatments included: 1 IU/mL LH (Luveris, Merck, Kenilworth, NJ, USA), 1 IU/mL hCG (Pregnyl, Merck, Kenilworth, NJ, USA), and vehicle (H_2_O + 0.01% BSA).

### 4.3. Western Blotting and Immunohistochemistry

Tissues and cell pellets were homogenized in lysis buffer (50 mM Tris pH 7.4, 100 mM NaCl, 2 mM EDTA, 1% Triton X-100, 0.5% NP-40) and diluted in SDS loading buffer (100 mM Tris, 25% Glycerol, 2% SDS, Bromophenol Blue, 10% β-mercaptoethanol) followed by heating at 95 °C for 5 min. 10 µg of each protein sample was loaded onto a 4–20% precast polyacrylamide gel (BioRad, Hercules, CA, USA, cat. no. 456-8096) and run for 1 h at 100 V to separate proteins. For detection of serum proteins, a Pierce Albumin/IgG Removal kit (Thermo Fisher Scientific, Waltham, MA, USA, cat. no. 89875) was used to decrease the abundant albumin and IgG components before loading of the samples onto the gel. After gel electrophoresis proteins were transferred to a polyvinylidene difluoride (PVDF) membrane (BioRad, Hercules, CA, USA, cat. no. 1620174) using a wet blot apparatus. Membranes were blocked for 1 h in Tris Buffered Saline (TBS) containing 5% nonfat dry milk before incubation with primary antibodies at 4 °C overnight and with secondary antibodies at RT for 1 h, both diluted in TBS containing 1% nonfat dry milk and 0.1% Tween 20. Membranes were developed using enhanced chemiluminescence (ECL) (Super Signal West Femto Maximum Sensitive Substrate, Thermo Fisher Scientific, Waltham, MA, USA, cat. no. 34095) and a chemiDoc MP imaging system (BioRad, Hercules, CA, USA, cat. no. 17001402) was used for photodetection. Immunohistochemical staining was performed as previously described with small modifications [42]. Briefly, sections were deparaffinized and rehydrated before antigen retrieval using a pressure cooker with TEG buffer. Sections were incubated with primary antibodies diluted in 5% BSA horse serum and later HRP secondary antibodies (Vector Laboratories, Burlingame, CA, USA). For negative controls serial sections were processed with the primary antibody replaced by dilution buffer alone. Counterstaining was performed with Mayer’s hematoxylin. All sections were scanned on a NanoZoomer 2.0 HT (Hamamatsu Photonics, Herrsching am Ammersee, Germany) and analyzed using the NDPview version 1.2.36 software (Hamamatsu Photonics, Herrsching am Ammersee, Germany). Primary antibodies are listed in Appendix A. LHCGR antibodies are generally of poor quality but the specificity of the LHR029 antibody has previously been validated in LHCGR or mock transfected cell lines [43]. Moreover, staining with LHR029 and LHRsc revealed expression in the membrane/cytoplasm of testicular Leydig cells in NT thus validating the specificity of the antibodies in human tissue.

### 4.4. Measurements of LHCGR in Serum

Serum samples were collected from 4 patients with GCNIS and 216 patients with TGCT prior to orchiectomy during their visit related to sperm freezing, and all samples were analyzed for LHCGR and routine tumor markers. Serum samples from 23 patients with non-seminoma (11/23 patients with relapse) and 20 patients with seminoma (9/20 patients with relapse) were obtained post orchiectomy and were only included if the men were followed longitudinally (minimum 3 samples/person) until relapse or the end of the 5 year follow-up period. Serum samples from 148 normal men, 297 infertile men, 5 men with Leydig cell tumor, and 4 men with infertility and Sertoli cell only pattern in testis biopsy were used for comparison. All men were included after approval from the regional ethical committee (H-17004362, KF 01 2006-3472). Measurements of LHCGR in serum were performed using a LHCGR ELISA (Novel Biomarkers Catalyst Lab (NBCL), Nijmegen, Netherlands and Origin Biomarkers, Hertfordshire, UK). The LHCGR ELISA has a limit of detection of 0.01 pmol/mL and an upper detection level of 15.55 pmol/mL and the validity of the assay has been described previously [43].

### 4.5. Establishment of Xenograft Mouse Models

All animal procedures were approved by the Danish Animal Experiments Inspectorate (Copenhagen, Denmark, License no. 2012-15-2934-00051). 6–8 weeks old NMRI nude mice (Fox1nu) were obtained from Taconic Europe. Mice were housed and interventions were performed at Pipeline Biotech in a designated pathogen-free area. European standard cages type 2 were used, and mice were fed Altromin 1324 ad libitum (Altromin, Lage, Germany). TCam2 and NTera2 cells were grown under standard conditions and plated in 175-cm2 flasks, and medium was changed every 48 h. Cells were injected subcutaneously into the right and left flank of male nude mice. Body weight was measured once a week and tumor volume three times a week from the time of inoculation until termination. Tumor volumes were calculated from two tumor diameter measurements using a Vernier caliper: tumor volume = L × W × 1/2W. When the tumors reached about 150 mm^3^ in size (day 17 and 35 for NTera2 and TCam2 xenografts, respectively), mice were randomized in different treatment groups (9 or 10 mice in each group for TCam2 and NTera2 xenografts, respectively) and test articles were administered. Mice were treated with three-times the usual adult dose 3 IE/kg LH 5 times weekly (Merck, Kenilworth, NJ, USA), 1000 IE/kg hCG three times weekly (Merck, Kenilworth, NJ, USA), 6 mg/kg Cisplatin once (Accord Healthcare Limited, Durham, NC, USA) or vehicle (PBS) three times weekly (Biological Industries, Cromwell, CT, USA). The treatment groups with Cisplatin and vehicle were also used in a separate study to reduce the total number of animals [44]. Mice were treated up to 14 days (NTera2 terminated at day 28 and TCam2 terminated at day 49) or until the humane endpoint of tumor size was reached (864 mm^3^, 12 mm diameter). After termination tumors, tissues and serum were isolated.

### 4.6. Statistical Analysis

Analysis of molecular data was performed using GraphPad Prism Software. Data are presented as mean ± SEM. Data from human cohorts were analyzed after stratification according to the histological subtype of the tumor and mean was calculated. Student’s *t*-test or ANOVA analysis followed by Dunnett’s test was used for comparison between groups, as appropriate. Human cohorts were analyzed with SPSS. *p* < 0.05 was considered statistically significant.

## 5. Conclusions

To sum up, this study shows that the expression of LHCGR in few GCNIS and a fraction of seminoma cells may be important for tumor growth if the stimulatory effects on proliferation of the TCam2 cells can be extrapolated into the clinical setting. The proposed direct stimulatory effect is supported by GCNIS and seminoma development in two patients with activating mutations in LHCGR, the release of LHCGR from seminoma cells, and the positive link between serum LHCGR and LDH and tumor burden in seminoma patients. 

## Figures and Tables

**Figure 1 cancers-12-01358-f001:**
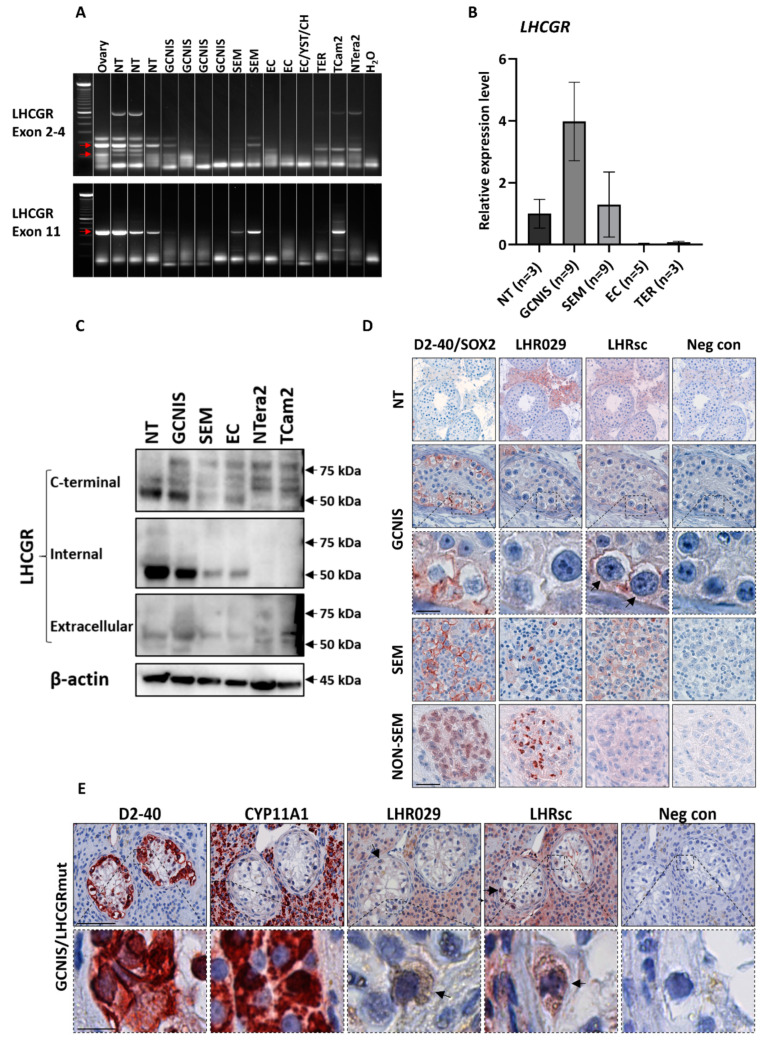
Expression of LHCGR in GCNIS and TGCTs. Expression of LHCGR in human testis cancer tissue and cell lines was analyzed by RT-PCR (**A**), qPCR (**B**), western blot (**C**), and IHC (**D**,**E**). (**A**) *LHCGR* mRNA expression analyzed by RT-PCR using two primer sets targeting exon 2-4 or exon 11. (**B**) *LHCGR* mRNA expression analyzed by qPCR using primer set targeting exon 11 and normalized to *RPS20*. (**C**) LHCGR protein expression analyzed by western blot using β-actin as loading control. (**D**) Localization of LHCGR investigated by IHC in TGCT tissue samples. D2-40 was used to mark GCNIS and seminoma cells while SOX2 was used to mark embryonal carcinoma cells in non-seminoma. Arrows indicate expression in GCNIS cells. Scale bar corresponds to 50 µm for all images except the higher magnification of GCNIS cells, in which the scale bar represents 12.5 µm for all images. (**E**) Localization of LHCGR in GCNIS tubules of a patient with a constitutively active mutation in LHCGR. D2-40 was used to mark GCNIS cells and CYP11A1 was used to mark Leydig cells. Arrows indicate expression in GCNIS cells. Scale bar in upper panel corresponds to 50 µm (all images), while scale bar in lower panel corresponds to 12.5 µm (all images). Abbreviations: CH, choriocarcinoma; EC, embryonal carcinoma; GCNIS, germ cell neoplasia in situ; NT, normal testis; SEM, seminoma; TER, teratoma; YST, yolk sac tumor.

**Figure 2 cancers-12-01358-f002:**
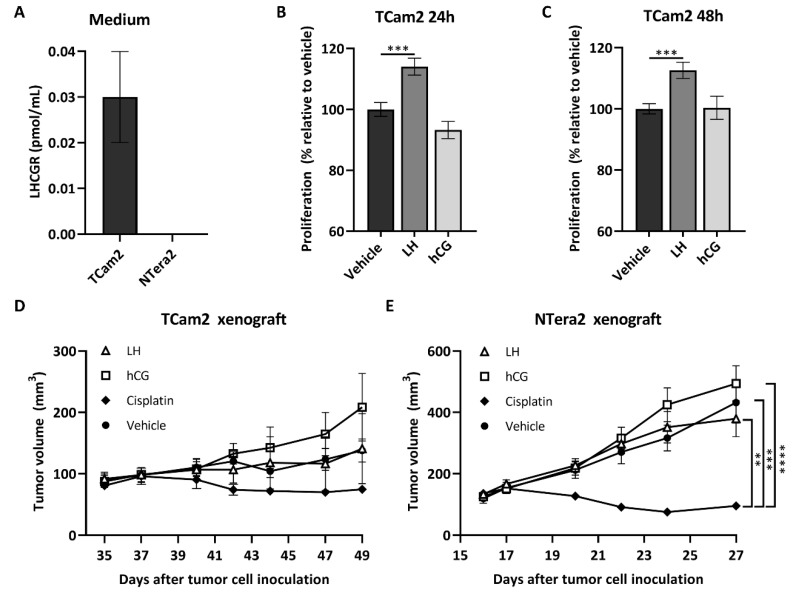
Stimulation of LHCGR increases proliferation of a seminoma-derived cell line. (**A**) LHCGR in cell culture medium from TCam2 and NTera2 cells measured by ELISA. TCam2 and NTera2 cells were grown until 85% confluency, the medium was changed, and the cells were cultured for 2–4 h before medium was harvested for analysis. Values represent mean ± SEM. The experiment was conducted in duplicates. (**B**,**C**) Cell proliferation of the seminoma-derived cell line TCam2 was analyzed by BrdU incorporation assay after 24 and 48 h of treatment with 1 IU/mL LH or hCG. Values represent mean ± SEM. All experiments are conducted in biological triplicates. *** *p* < 0.001 compared with vehicle treatment. (**D**,**E**) NTera2 or TCam2 cells were inoculated onto the flank of nude mice. At an average tumor volume of approximately 150 mm^3^, mice were randomized into different treatment groups and injected i.p. with LH (3 IU/kg 2 times/week), hCG (1000 IU/kg 3 times/week), cisplatin (6 mg/kg once), or vehicle (PBS) for 14 days. Tumor volume was measured 3 times/week and body weight determined once a week from the time of inoculation until termination. Values represent mean ± SEM for *n* = 10 (**D**) and *n* = 9 (**E**). ** *p* < 0.01, *** *p* < 0.001, and **** *p* < 0.0001. Abbreviations: hCG, human chorionic gonadotropin; LH, luteinizing hormone; LHCGR, luteinizing hormone/choriogonadotropin receptor.

**Figure 3 cancers-12-01358-f003:**
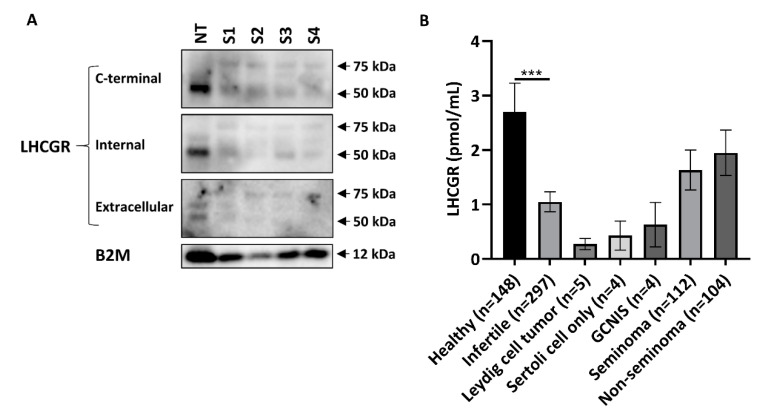
LHCGR in serum from patients with TGCTs. (**A**) Western blot analysis of serum LHCGR. NT: normal testis tissue from a patient with GCNIS. S1: serum from a healthy man with low LHCGR. S2: serum from a healthy man with high LHCGR. S3: serum from a patient with seminoma and high serum LHCGR. S4: serum from a patient with non-seminoma and high serum LHCGR. Serum levels of LHCGR were measured by a validated ELISA platform. B2M was used as loading control. (**B**) LHCGR serum levels analyzed using a LHCGR ELISA in men with non-malignant conditions and TGCTs. Values represent mean ± SEM. *** *p* < 0.001 compared with healthy controls. Abbreviations: B2M, beta-2-microglobulin; GCNIS, germ cell neoplasia in situ; LHCGR, luteinizing hormone/choriogonadotropin receptor.

**Figure 4 cancers-12-01358-f004:**
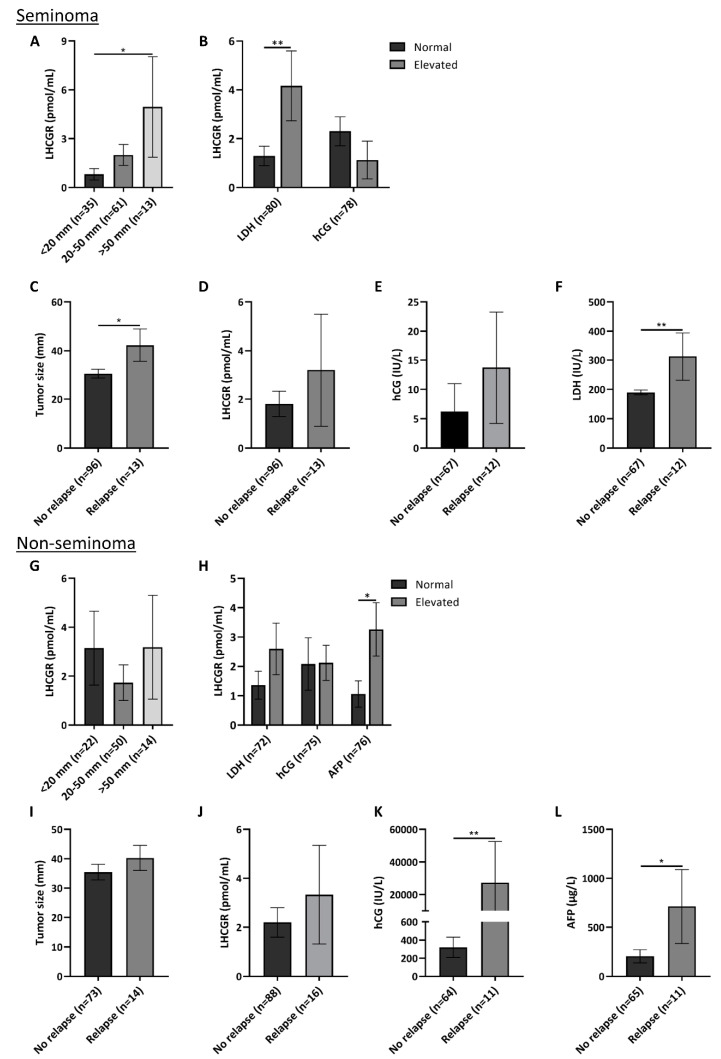
Serum LHCGR is associated with LDH and tumor size in patients with seminoma. (**A**,**B**) LHCGR serum levels were analyzed in patients with seminoma stratified according to tumor size (**A**) and serum LDH (<205 vs. ≥205 U/L) and hCG (<3 vs. ≥3 IU/L) levels (**B**). (**C**–**F**) Tumor size (**C**) and serum levels of LHCGR (**D**), hCG (**E**), and LDH (**F**) were measured in patients with seminoma stratified into groups depending on whether they had experienced a relapse or not. (**G**,**H**) LHCGR serum levels were analyzed in patients with non-seminoma stratified according to tumor size (**G**) and serum LDH (<205 vs. ≥205 U/L), hCG (<3 vs. ≥3 IU/L) and AFP (<12 vs. ≥12 × 10³ IU/L) levels (**H**). (**I**,**L**) Tumor size (**I**) and serum levels of LHCGR (**J**), hCG (**K**), and AFP (**L**) were measured in patients with non-seminoma stratified into groups depending on whether they had experienced a relapse or not. Values represent mean ± SEM. * *p* < 0.05, ** *p* < 0.01. Abbreviations: AFP, Alpha-fetoprotein; hCG, human chorionic gonadotropin; LDH, lactate dehydrogenase; LHCGR, luteinizing hormone/choriogonadotropin receptor.

**Figure 5 cancers-12-01358-f005:**
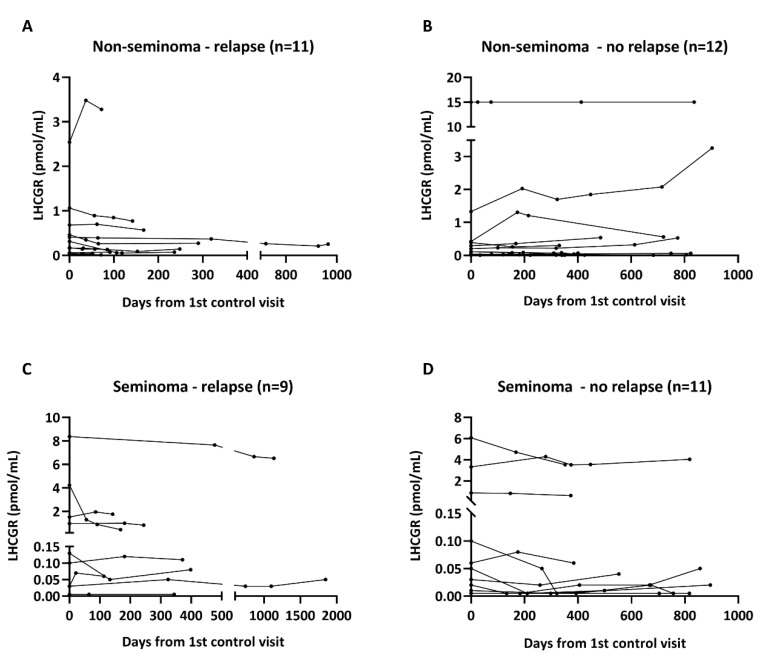
Serum LHCGR in TGCT patients with and without relapse. LHCGR serum levels were measured in patients with non-seminoma (**A**,**B**) and seminoma (**C**,**D**) who were followed longitudinally until they experienced a relapse or were released from the follow up period. Abbreviations: LHCGR, luteinizing hormone/choriogonadotropin receptor.

**Table 1 cancers-12-01358-t001:** Detection of LHCGR by different techniques.

	RT-PCR	qPCR	IHC	WB
GCNIS	−/+	++	−/+	++
Seminoma	+/++	++	+/++	+
Embryonal carcinoma	−	−	−/+	+
NTera2	−	NA	NA	+
TCam2	+	NA	+	+

IHC was conducted using two different LHCGR antibodies, while WB was conducted using three different antibodies and the results are the average expression of all antibodies tested. Expression was classified according to an arbitrary semi-quantitative reference scale depending on overall expression of LHCGR with the specific methodology: ++, strong; +, weak; −, not detectable; NA, not available. Abbreviations: GCNIS, germ cell neoplasia in situ; IHC, immunohistochemistry; LHCGR, luteinizing hormone/choriogonadotropin receptor; RT-PCR, reverse transcription polymerase chain reaction; WB, western blot; qPCR, quantitative polymerase chain reaction.

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
