# Peer review of "Luteinizing Hormone Receptor Is Expressed in Testicular Germ Cell Tumors: Possible Implications for Tumor Growth and Prognosis"

_cancers, 2020, doi:10.3390/cancers12061358_

Round 1

Reviewer 1 Report

In their manuscript Lorenzen et al investigate the expression of LHCGR in human testicular germ cell cancers, using a range of techniques (in vitro studies, IHC, WB, ELISA, RT-PCR, qPCR, in vivo xenografting, correlation analysis), analysing in vitro/in vivo challenged cell lines as well as a big cohort of patient samples, both tissue as well as serum specimens.

  1. Overall, from the reviewer´s point of view, data summarised and presented is not stringently coherent with different techniques indicating different results for the different groups. Results should be described more precisely. Were samples studied by qPCR, WB, IHC etc. matched or were individual samples analysed by one or the other technique? This aspect should be clarified, mentioned and discussed.
  2. To provide a proper overview and summary of the different data sets, it would be of value to add a table summarising all findings for the different groups generated by the different techniques. Based on that, by looking at the cumulative data, how do you judge the differences in outcomes provided by application of techniques, antibodies, primers etc? Which statement can be supported by several data sets and which results appear to be singular events, not replicable by different methods?
  3. The discussion should be more precise in giving a final overall statement and summary of results and interpretations rather than listing results and interpretations individually. The entitiy of results provided and generated by application of several techniques should at last be put together into a more global picture describing the final study meaning and critically judging the different techniques (individual antibodies etc) that were applied.
  4. Some legends could be extended to provide further information on e.g. n-numbers or treatment doses/frequencies that are otherwise only mentioned in the materials and methods section.
  5. Overall, mean +/- SD should be presented rather than SEM and also mentioned in the figure legends, since SD illustrates variability within a group.
  1. 1 A: the PCR product i.e. band specific for the primer pair targeting exon 11 should also be indicated by an arrow. It should also be indicated in graph B, that this primer pair was used for the qPCR analysis. It is unclear to the reviewer why data for primer pair targeting exons 2 and 4 is also being shown – clearly, this primer pair is not very specific producing multipe products/bands. Judging by the gels, the results are not in line with the data generated by using primer pair targeting exon 11: exon 11 primer is NT +ve, GCNIS –ve, SEM +ve, TCam2 +ve, NTera2 and others –ve; exon 2 + 4 primer is NT +ve, GCNIS unclear, SEM unclear, TCam2 and NTera2 unclear but similar. Also, lines could be included in the image to seperate groups from one another, facilitating interpretation of data. From the reviewer´s point of view and judging by the arrows, TER, TCam2 and NTera2 would be negative since the arrows point at different products. Please be more precise in the figures and more accurate in the interpretation of results.
  2. 1 B: Is there a difference between group GCNIS (usually used throughout the study) and NT/GCNIS as mentioned in this graph? Please specify. For completeness, qPCR data for TCam2 and NTera2 cells should also be included in the graph. Moreover, qPCR data does not seem to match RT-PCR results; e.g. GCNIS gel –ve, qPCR clearly +ve. Please comment and explain. For the materials and methods section, please indicate how much total RNA was transcribed into cDNA (1 µl cDNA per reaction is no quantitative value).
  3. Considering Fig. 1 A + B, the authors summarise the results by mentioning that „both RT-PCR and qPCR showed expression of LHCGR in NT, GCNIS and SEM…“. From the reviewer´s point of view, this comment is very generalised and a more precise summary of results should be provided since data is actually not as consistent as mentioned (see comment above).
  4. The authors mention the CT values for LHCGR to support their findings. From the reviewers point of view this information may only be of value if further information is being provided such as, in comparison, the CT values of the endogenous control (as a reference), the total number of cycles that were run as well as the cut-off CT value at which the product is considered to be „absent“. This is especially important considering that the authors mention that NT, GCNIS, SEM and TCam2 are +ve (CT values 26 – 29), whereas NTera2 cells with CT values between 30 to 33, as little as 1 cycle apart, are considered negative. Please comment.
  5. The authors verify their data by performing WB using three different antibodies targeting different regions of LHCGR (C-terminal, internal, extracellular) and postulate that a clear band around 50 kDa is visible in all tissues (NT, GCNIS; EC, SEM). Additionally, ll 78 – 79 „all antibodies revealed similar results with a clear band around 50 kDa in all tissues …“: This comment once again appears quite generalised, especially considering the antibody „extracellular“, which presents with bands being quite fluctuant. Especially in WB „extracellular“, the arrow indicating a band at 50 kDa appears to be (i) not put at the right site or (ii) indicates that in fact, only NTera2 and TCam2 cells present a positive band at 50 KDa (whereas all other samples present a band slightly above the 50 kDa indicator).
  6. What is the expected molecular weight for each of these antibodies used in WB? What antibodies (product no, supplier) were used? The details are not to be found in the materials and methods section. Please add. What specimen was used as a positive control and how were the antibodies validated in general? Also, please mention in the end which antibody appears to be the most reliable one to you since again data is not stringently coherent (e.g. NTera2 cells which in most data sets appear to be LHCGR –ve present a band in WB that was indicated by the authors to represent LHCGR protein; on the other hand, unlike qPCR data, TCam2 cells appear to be rather LHCGR –ve based on WB data).
  7. Additionally, how much protein was loaded for Western Blot analyses? Please add this information to the materials and methods section.
  8. 1 D: two antibodies were used for immunohistochemistry. What antibodies (product no, supplier) were used in specific? The details are not to be found in the materials and methods section. Please add. How were these antibodies validated? Were the same antibodies used in the WB analyses? Both antibodies are targeting the extracellular domain of LHCGR but present different staining patterns – how do you explain the differences? How do you „rank“ the antibodies? To the reviewer, LHRsc seems to also stain spermatogenic cells. Please comment.
  9. 1 E: staining for LHR029 should be added for completeness.
  10. The origin of TCam2 cells is mentioned in the abstract already, whereas the origin of NTera2 cells is only mentioned very late (in the materials and methods section at the end of the manuscript only). This info should be provided earlier to explain the use of these two different cell lines in the study´s setting.
  11. Ll 106 – 107, 109 – 110: what is this referring to? There appears to be no data included in this manuscript in which cells were stained with OCT4.
  12. How come for in vitro analyses, similar doses were used for both LH and hCG, whereas in the xenograft in vivo model, substantially different doses and application regimes were applied. Please explain the rationale behind that and explain how treatment doses for both substances in in vitro vs in vivo studies were defined, i.e. dose-response curves should be included.
  13. How do you explain the difference between in vitro and in vivo data? E.g. NTera2 cells treated with LH and hCG in vitro did not respond at all, in vitro TCam2 cells responded to LH only, whereas under in vivo circumstances, both TCam2 and NTera2 cells showed a response to hCG only. How do you put this data together? What does it mean, imply or suggest?
  14. 2A: please indicate in the graph/legend, at which timepoint the medium was harvested for analysis and for how long/up to which confluency cells were grown.
  15. 2 B and C: please use the same layout.
  16. 3 A: patient sera applied to WB - how were these patients identified to have high or low LHCGR levels? Bands are very hard to distinguish and to verify. Fig. B: n numbers should be included in graph and/or legend.
  17. 4: what is the cut-off value for normal vs elevated LDH, hCG or AFP? Could this information be added to the graph, similar to what was done in Fig. 4 A?

Reviewer 2 Report

In this study the authors want to evaluate LHCGR may be used as a novel biomarker for seminomas by investigating the expression of LHCGR in TGCT patients and mouse mode. The applications on investigating the expression of LHCGR in GCNIS, TGCTs and TGCT-derived cells lines and the effects of LHCGR activation on proliferation of testicular cancer cell lines in vitro and in two tumor xenograft mouse models are appreciated. Nevertheless, the conclusion is hard to comprehend and the composition of manuscript should be improved.

The present status of biomarker for seminomas should be added in Introduction.

All the abbreviated word should be together in one place but not in each legend.

The quality of images of Figure 1D is very low and it is not possible to identify the localization of LHCGR. And the scale bar in each image should be confirmed.

The explanations about antibodies LHR029 and LHRsc should be removed to Material & Methods section. And as the author wrote “LHCGR antibodies are generally of poor quality but the specificity of the LHR029 antibody has previously been validated in LHCGR or mock transfected cell lines”, the IHC in GCNIS tubules should include the LHR029 but not only LHRsc.

The explanations of the black arrows in Figure 1D and 1E are deficient.

The number of patients in the analyzed LHCGR serum levels should be added in Figure 3B.

The discussion included the summarize the results and formulate an opinion about this study. The authors should compare the previous studies according data and the result of each Figure in this study. Furthermore, to clarify the conclusion, a schematic view is necessary.

The references should not be cited in Results usually (ref16 and 19; line 103-115).

Round 2

Reviewer 1 Report

The authors have adequately addressed all comments raised, have clarified and specified the study´s specific results and have put the data into a bigger picture.  

Reviewer 2 Report

In the revised manuscript, the authors have added a schematic view in Table 1 and improved the quality of Figure 1D and 1E partially. But the localization of LHCGR in GCNIS cells indicated by arrows are also not possible to identify clearly. It is better adding enlarged part of the LHCGR expressed in GCNIS cells.
